# Reweighted Flow Matching via Unbalanced Optimal Transport for Long-tailed Generation

## Abstract

Flow matching has recently emerged as a powerful framework for continuous-time generative modeling. However, when applied to long-tailed distributions, standard flow matching suffers from majority bias, oversampling majority modes while generating minority modes with low fidelity. In this work, we propose UOT-Reweighted Flow Matching (UOT-RFM), which leverages Unbalanced Optimal Transport (UOT) to estimate an unsupervised majority score for each target data. Using this score, we correct bias via inverse weighting and introduce higher-order corrections ($k > 1$) to further emphasize minority modes. We establish a bias correction theorem, showing that first-order weighting exactly recovers the target distribution. We show that UOT-RFM outperforms existing flow-matching baselines by improving diversity and fidelity on synthetic long-tail data and CIFAR-10-LT.

## 1 Introduction

Generative modeling addresses the problem of approximating a target data distribution. Deep generative models have achieved remarkable progress in recent years, such as GANs [1, 15], optimal transport maps [7, 9, 28], and diffusion models [17, 30]. Among them, flow matching models [22] have emerged as a promising approach for continuous-time generative models. Flow matching learns a continuous normalizing flow [4], i.e., a vector field describing the dynamics between an initial prior distribution and the target distribution, while avoiding costly numerical likelihood estimation. Flow matching model is trained through regression to conditional vector field, constructed from conditional probability path between prior and target samples.

Despite these advances, flow matching models remain vulnerable to majority bias when trained on long-tailed distributions. In many real-world datasets, data often follow long-tailed or imbalanced distributions, where a few classes dominate while minority classes are severely underrepresented [3, 27, 33]. In such cases, standard flow matching tends to overfit the majority mode due to its regression-based learning nature, while undersampling or exhibiting low fidelity for the minority mode. This limitation reduces diversity and degrades the quality of samples from rare classes.

To overcome these challenges, we propose the flow matching model based on the Unbalanced Optimal Transport (UOT) [6, 21]. A key property of UOT is that it naturally produces a density ratio between the target distribution and the UOT marginal, which we call the **majority score**. Building on this, we propose *UOT-Reweighted Flow Matching (UOT-RFM)*, which corrects bias via inverse weighting and allows high-order corrections ($k > 1$) to further emphasize tail samples. Our method provides an unsupervised bias-correction mechanism and enhances coverage of long-tailed data. Our experiments on the CIFAR-10-LT benchmark demonstrate that our method outperforms existing flow matching baselines. Our contributions can be summarized as follows:

Submitted to 39th Conference on Neural Information Processing Systems (NeurIPS 2025). Do not distribute.

- We propose UOT-RFM, which leverages UOT couplings and the majority score for unsupervised bias correction.
- We establish a bias correction theorem, proving that first-order inverse weighting with the majority score recovers the true target distribution.
- Our experiments show that our method achieves improved performance on long-tailed data generation and offers a principled trade-off between majority and minority emphasis with higher-order correction.

## 2  Preliminaries

**Flow Matching**  Continuous Normalizing Flows (CNFs) [4, 22] model the dynamics of the probability densities through a *probability density path* $p(t, \mathbf{x}) : [0, 1] \times \mathbb{R}^d \mapsto \mathbb{R}_{\geq 0}$ which transports the initial or source distribution (e.g., Gaussian distribution) $p_0$ to the target data distribution $p_1$. Specifically, the CNF model is defined by the following Ordinary Differential Equation (ODE), governed by a parameterized vector field $\mathbf{v}^\theta : [0, 1] \times \mathbb{R}^d \mapsto \mathbb{R}^d$, i.e., $\frac{\mathrm{d}x}{\mathrm{d}t} = \mathbf{v}_t^\theta(\mathbf{x})$, where we use the notation $\mathbf{v}_t(\mathbf{x})$ interchangeably with $v(t, \mathbf{x})$. Then, the associated flow map $\phi_t(\mathbf{x})$ denotes the solution of this ODE with initial condition $\phi_0(\mathbf{x}) = \mathbf{x}$ and the density at time $t$ is given by $p_t = (\phi_t)_\# p_0$.

Lipman et al. [22] proposed *flow matching*, a scalable method for training CNFs. The idea is to train the CNF by minimizing a regression loss $\mathcal{L}_{\mathrm{FM}}(\boldsymbol{\theta})$ between the parameterized vector field $v_t^\theta$ and the ground-truth vector field $u_t$ that generates the probability path $p_t$. However, a major challenge is that the marginal ground-truth vector field $u_t$ is intractable.

$$\mathcal{L}_{\mathrm{FM}}(\boldsymbol{\theta}) = \mathbb{E}_{t \sim \mathcal{U}, \mathbf{x}_t \sim p_t(\mathbf{x}_t)} \| v_\theta(t, \mathbf{x}_t) - u_t(\mathbf{x}_t) \|_2^2. \tag{1}$$

To overcome this, the flow matching [22, 31] introduces a conditional flow matching. Instead of matching $u_t$, the model is trained against the tractable *conditional vector field* $u_t(\mathbf{x}_t | \mathbf{z})$, which generates a *conditional probability path* $p_t(\mathbf{x}_t | \mathbf{z})$, where $\mathbf{z}$ denotes sample pairs $(\mathbf{x}_0, \mathbf{x}_1)$. The sample pairs $(\mathbf{x}_0, \mathbf{x}_1)$ follow the joint distribution (couplings) of $\pi(\mathbf{z}) = \pi(\mathbf{x}_0, \mathbf{x}_1)$. The training objectives are given by

$$\mathcal{L}_{\mathrm{CFM}}(\boldsymbol{\theta}) = \mathbb{E}_{t \sim \mathcal{U}, \mathbf{z} \sim \pi(\mathbf{z}), \mathbf{x}_t \sim p_t(\mathbf{x}_t | \mathbf{z})} \| v_\theta(t, \mathbf{x}_t) - u_{t|\mathbf{z}}(\mathbf{x}_t | \mathbf{z}) \|_2^2. \tag{2}$$

CFM replaces the intractable marginal vector field with a tractable conditional one based on couplings. In particular, the conditional probability path $p_t(\mathbf{x}_t | \mathbf{z})$ and the associated conditional vector field $u_t(\mathbf{x}_t | \mathbf{z})$ can be defined as follows [31]:

$$p_t(\mathbf{x}_t \mid \mathbf{z}) = \mathcal{N}\left(\mathbf{x} \mid t\mathbf{x}_1 + (1 - t)\mathbf{x}_0 \mid \sigma^2\right), \quad u_t(\mathbf{x}_t \mid \mathbf{z}) = \mathbf{x}_1 - \mathbf{x}_0 \tag{3}$$

for some bandwidth hyperparameter $\sigma > 0$. In this case, the marginal probability path and the marginal vector field that generates this path are given by

$$p_t(\mathbf{x}_t) = \int p_t(\mathbf{x}_t \mid \mathbf{z}) \pi(\mathbf{z}) d\mathbf{z}, \quad u_t(\mathbf{x}_t) := \mathbb{E}_{\pi(\mathbf{z})} \frac{u_t(\mathbf{x} \mid \mathbf{z}) p_t(\mathbf{x} \mid \mathbf{z})}{p_t(\mathbf{x})} = \mathbb{E}_{p_t(\mathbf{z} | \mathbf{x}_t)} \left[ u_t(\mathbf{x}_t \mid \mathbf{z}) \right] \tag{4}$$

**Initial Coupling in Flow Matching**  A key component in the training flow matching model is the choice of initial sample couplings $\pi(\mathbf{z}) = \pi(\mathbf{x}_0, \mathbf{x}_1)$. **This coupling determines how the flow matching model is trained**, because the obtained model $v_t(\mathbf{x}_t) \approx u_t(\mathbf{x}_t)$ relies on aggregating the conditional vector field over paired samples $p_t(\mathbf{z} | \mathbf{x}_t)$ (Eq. 4). The original flow matching framework [22] employs an independent coupling between the source and target distributions. However, such independence often leads to curved trajectories, which arises from the mean-shift phenomenon due to the flow crossing problem [20, 24]. These curved trajectories result in increased numerical errors in ODE simulation and thereby high computational costs in sampling [23].

To improve couplings, recent works adopted the *Optimal Transport (OT)* approaches between minibatches [26, 31]. Note that the Kantorovich formulation of the Optimal Transport is given by

$$C_{ot}(\mu, \nu) := \inf_{\pi \in \Pi(\mu, \nu)} \left[ \int_{\mathcal{X} \times \mathcal{Y}} c(\mathbf{x}, \mathbf{y}) d\pi(\mathbf{x}, \mathbf{y}) \right]. \tag{5}$$

Here, the optimal coupling $\pi^\star$ is defined as the minimizer of the transport cost $c(x, y)$ between empirical measures of minibatches from the source samples $\mathbf{x}_0$ and target samples $\mathbf{x}_1$. Alternatively, Rectified flow [23] proposed that leverages pretrained flow models to improve couplings. In this approach, the trajectories are iteratively refined using the previous model as the initial coupling, resulting in straighter paths.

## 3 Method

In this section, we present our model, called ***UOT-Reweighted Flow Matching (UOT-RFM)***, that addresses the majority bias of existing flow matching approaches on long-tailed distributions. Our model leverages minibatch Unbalanced Optimal Transport coupling, which naturally provides a ***majority score*** for each sample. Intuitively, we compensate for majority bias by over-correcting each target data utilizing this score. In Sec 3.1, we introduce the Uabalanced Optimal Transport problem. In Sec 3.2, we introduce our UOT-RFM model.

### 3.1 Unbalanced Optimal Transport

We introduce the *Unbalanced Optimal Transport* problem [6, 21] and its key properties, which will be leveraged in our approach. The standard OT problem (Eq. 5) enforces *exact transport* between the source and target distributions, i.e., $\pi_0 = \mu, \pi_1 = \nu$. However, this strict marginal constraint makes OT sensitive to outliers [2, 7, 14, 29]. To address these issues, the *Unbalanced Optimal Transport problem* relaxes this constraint and introduces the divergence penalties on the marginal distributions.

$$C_{uot}(\mu, \nu) = \inf_{\pi \in \mathcal{M}_+(\mathcal{X} \times \mathcal{Y})} \left[ \int_{\mathcal{X} \times \mathcal{Y}} c(x, y) d\pi(x, y) + \tau_1 D_{\Psi_1}(\pi_0 \| \mu) + \tau_2 D_{\Psi_2}(\pi_1 \| \nu) \right], \quad (6)$$

where we assume $c(x, y) = \frac{1}{2}\|x - y\|_2^2$ and $\tau_1, \tau_2 > 0$ control the strength of the marginal matching penalties. Here, $\mathcal{M}_+(\mathcal{X} \times \mathcal{Y})$ indicates the set of positive Radon measures on $\mathcal{X} \times \mathcal{Y}$. The terms $D_{\Psi_1}(\pi_0 \| \mu)$ and $D_{\Psi_2}(\pi_1 \| \nu)$ are two $f$-divergences that penalizes deviations of the marginals $\pi_0, \pi_1$ from the source $\mu$ and target $\nu$, respectively.

Therefore, the optimal UOT coupling $\pi^u$ softly matches $\mu$ and $\nu$, i.e., $\pi_0^u \approx \mu$ and $\pi_1^u \approx \nu$. Moreover, the UOT problem can represent exact matching of one marginal by setting the divergence penalty appropriately. Specifically, if $\Psi_i$ is the convex indicator function $\iota$ at $\{1\}$, then $D_\iota(\pi_i \| \eta) = 0$ if $\pi_i = \eta$ a.s., and $\infty$ otherwise. For example, if $\Psi_1 = \iota$, we obtain at the **source-fixed UOT problem** where $\pi_0^u = \mu$ and $\pi_1^u \approx \nu$.

### 3.2 Proposed Method

**Majority Score**  Our method leverages the **mini-batch UOT coupling** $\pi^u$ and the resulting **majority score** $s_\tau(\cdot) = (d\pi_1^u/d\nu)$. This score is utilized to address the majority oversampling bias of flow matching models on long-tailed distributions by inversely reweighting each target sample. Intuitively, the optimal UOT coupling $\pi^u$ exhibits distribution error whenever a small increase in $D_\Psi$ leads to a large decrease in transport cost $c(x, y)$ (Eq. 6). As a result, $\pi^u$ prioritizes matching the majority modes, while down-weighting outlier modes with small mass and large cost. This property explains the robustness of UOT to outliers, as the UOT effectively reduces the influence of outliers [2, 7, 29].

Based on this property, we define the ***majority score*** $s_\tau(\cdot) = (d\pi_1^u/d\nu)$ as the density ratio in the target space under the source-fixed UOT problem.

$$s_\tau(y) := \frac{d\pi_1^{u,\star}}{d\nu}(y) > 0 \quad (7)$$

Here, $\tau_1$ is irrelevant, so we simply set $\tau = \tau_2$. Intuitively, the majority score measures how strongly each target sample is emphasized by the UOT coupling. $s_\tau > 1$ indicates emphasized majority samples, while $s_\tau \ll 1$ correspond to down-weighted outlier samples. Importantly, this weighting is entirely **unsupervised**, arising from the intrinsic geometry of probability distributions (see Appendix A for details).

**Proposed Method**  Our corrected conditional flow matching objective with correction order $k \geq 1$ is defined as follows (Algorithm 1):

$$\mathcal{L}_{\text{ours,k}}(\boldsymbol{\theta}) = \mathbb{E}_{t \sim \mathcal{U}, \mathbf{z} \sim \pi^u(\mathbf{z}), \mathbf{x}_t \sim p_t(\mathbf{x}_t | \mathbf{z})} \left[ s_\tau(\mathbf{x}_1)^{-k} \| v_\theta(t, \mathbf{x}_t) - u_{t|\mathbf{z}}(\mathbf{x}_t | \mathbf{z}) \|_2^2 \right]. \quad (8)$$

where the conditioning variable $\mathbf{z} = (\mathbf{x}_0, \mathbf{x}_1)$. Compared with standard flow matching (Eq. 3), our formulation employs the UOT coupling $\pi^u$ for pairing $\mathbf{z}$ and introduce an additional weighting factor $s_\tau(\mathbf{x}_1)^{-k}$ that rebalances majority and minority samples. Our method is motivated by the following bias correction theorem (see Appendix B for formal statements and proof):

Table 1: **Quantitative results on CIFAR-10-LT**. In both settings, the training data is long-tailed. The test data is either long-tailed (*LT→LT*) or balanced (*LT→Balanced*).

| Model | LT→LT | | | | LT→Balanced | | | |
|---|---|---|---|---|---|---|---|---|
| | FID ($\downarrow$) | Prec ($\uparrow$) | Recall ($\uparrow$) | F1 ($\uparrow$) | FID ($\downarrow$) | Prec ($\uparrow$) | Recall ($\uparrow$) | F1 ($\uparrow$) |
| I-CFM | 14.57 | 0.67 | 0.28 | 0.39 | 25.46 | 0.60 | 0.22 | 0.32 |
| OT-CFM | 17.31 | **0.71** | 0.24 | 0.36 | 27.51 | **0.63** | 0.16 | 0.26 |
| UOT-CFM | 14.25 | 0.67 | 0.29 | 0.41 | 24.94 | 0.59 | 0.23 | 0.33 |
| Ours | **11.03** | 0.61 | **0.41** | **0.49** | **24.06** | 0.55 | **0.38** | **0.45** |

Table 2: **Ablation study on the correction order** $k$ when $\tau = 2.0$. Reported values are FID scores.

| Training→Test | Correction order $k$ | | | | | | UOT-CFM ($k = 0$) |
|---|---|---|---|---|---|---|---|
| | 1.0 | 2.0 | 4.0 | 6.0 | 8.0 | 10.0 | |
| LT→LT | 13.77 | 13.41 | 12.42 | 11.72 | 11.37 | **11.04** | 14.25 |
| LT→Balanced | 25.02 | 24.60 | **24.54** | 24.70 | 24.76 | 25.35 | 24.94 |

**Theorem 3.1** (Informal). *Flow matching with UOT coupling generates a biased distribution $p_1 = \pi_{\tau,1}^u \neq \nu$, which overweights majority modes. UOT-RFM corrects this bias by reweighting with the majority score: with correction order $k$, it generates $p_1 \propto s_\tau^{-k} \pi_{\tau,1}^u$. In particular, $k = 1$ exactly recovers the true target distribution $\nu$.*

Theorem 3.1 shows that when training a flow matching model with UOT coupling (UOT-CFM, [8]), the generated distribution $p_1$ is biased, i.e., $p_1 = \pi_1^u \neq \nu$. In particular, the distribution $\pi_1^u$ magnifies the majority modes while suppressing the tail modes. This bias can be corrected by applying inverse weighting with the majority score $s_\tau$. Our method extends this idea with **over-correction** ($k > 1$), further emphasizing tail samples with $s_\tau(\cdot) < 1$. Unlike OT-CFM [31], which relies on mini-batch OT coupling, our approach provides an **unsupervised estimate of the majority score**, without requiring class labels [27, 33].

## 4 Experiments

We evaluate our model on long-tailed distributions. In each experiment, our model is compared with several flow matching baselines: independent coupling (I-CFM), OT coupling (OT-CFM, [25, 31]), and UOT coupling (UOT-CFM, [8]).

**Long-Tailed CIFAR-10**  We evaluate our model on CIFAR-10 under two settings, using long-tailed CIFAR-10 (CIFAR-10-LT) [3] as the training data in both cases. In the first setting (LT→LT), the test set is also CIFAR-10-LT, assessing how well each model fits the long-tailed distribution. In the second setting (LT→Balanced), the test set is the original balanced CIFAR-10 [18], evaluating whether a model trained on imbalanced data can recover the balanced distribution. This evaluation setup is often adopted in supervised long-tailed learning [27, 33]. Performance is measured using FID [16], Precision, Recall, and F1-score [19].

Table 1 reports the quantitative results (see Appendix E for qualitative examples). In both settings, our model outperforms all flow matching baselines. In particular, our model achieves significant improvement in the Recall metric, demonstrating improved coverage of minority modes. Although OT-CFM achieves the best precision metric, our model achieves the best F1-score, which comprehensively evaluates the Precision and Recall metrics. Moreover, note that the additional training cost is minimal: UOT-RFM requires only about 7% more time than OT-CFM.

**Correction Order**  We conduct an ablation study on the correction order $k$ to examine its impact on performance. Interestingly, the best FID scores are achieved when $k > 1$, rather than with the exact correction $k = 1$. Moreover, compared to UOT-CFM (UOT-RFM without correction), introducing correction generally improves FID scores. Overall, UOT-RFM remains robust to correction order, outperforming other baseline models for all moderate correction orders $2 \leq k \leq 8$.

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

## A Unbalanced Optimal Transport

The classical OT problem assumes an exact transport between two distributions $\mu$ and $\nu$, i.e., $\pi_0 = \mu, \pi_1 = \nu$. However, this exact matching constraint results in sensitivity to outliers [2, 29] and vulnerability to class imbalance in the OT problem [8]. To mitigate this issue, a new variation of the OT problem is introduced, called *Unbalanced Optimal Transport (UOT)* [6, 21]. Formally, the UOT problem is expressed as follows:

$$C_{uot}(\mu, \nu) = \inf_{\pi \in \mathcal{M}_+(\mathcal{X} \times \mathcal{Y})} \left[ \iint_{\mathcal{X} \times \mathcal{Y}} c(x, y) d\pi(x, y) + D_{\Psi_1}(\pi_0 | \mu) + D_{\Psi_2}(\pi_1 | \nu) \right], \quad (9)$$

where $\mathcal{M}_+(\mathcal{X} \times \mathcal{Y})$ denotes the set of positive Radon measures on $\mathcal{X} \times \mathcal{Y}$. $D_{\Psi_1}$ and $D_{\Psi_2}$ represents two $f$-divergences generated by convex functions $\Psi_i$, and are defined as $D_{\Psi_i}(\pi_j | \eta) = \int \Psi_i \left( \frac{d\pi_j(x)}{d\eta(x)} \right) d\eta(x)$. These $f$-divergences penalize the discrepancies between the marginal distributions $\pi_0, \pi_1$ and $\mu, \nu$, respectively. Hence, **in the UOT problem, the two marginal distributions are softly matched to** $\mu, \nu$, i.e., $\pi_0 \approx \mu$ and $\pi_1 \approx \nu$. Intuitively, the UOT problem can be seen as the OT problem between $\pi_0 \approx \mu$ and $\pi_1 \approx \nu$, rather than between the exact distributions $\mu$ and $\nu$ [7]. This flexibility offers robustness to outliers [2] and adaptability to class imbalance problem between $\mu$ and $\nu$ [8] to the UOT problem.

Similar to the standard OT problem, the UOT problem also admits a *dual formulation* [7, 13, 32]:

$$C_{uot}(\mu, \nu) = \sup_{u(x)+v(y) \le c(x,y)} \left[ \int_{\mathcal{X}} -\Psi_1^*(-u(x)) d\mu(x) + \int_{\mathcal{Y}} -\Psi_2^*(-v(y)) d\nu(y) \right], \quad (10)$$

with $u \in \mathcal{C}(\mathcal{X})$, $v \in \mathcal{C}(\mathcal{Y})$ where $\mathcal{C}$ denotes a set of continuous functions over its domain. Here, $f^*$ denotes the *convex conjugate* of $f$, i.e., $f^*(y) = \sup_{x \in \mathbb{R}}\{\langle x, y \rangle - f(x)\}$ for $f : \mathbb{R} \to [-\infty, \infty]$. Note that this dual problem conducts maximization over two continuous functions $u$ and $v$. This dual problem can be simplified into a *semi-dual* formulation by eliminating $u$ via the optimality condition:

$$C_{uot}(\mu, \nu) = \sup_{v \in \mathcal{C}} \left[ \int_{\mathcal{X}} -\Psi_1^*(-v^c(x))) \, d\mu(x) + \int_{\mathcal{Y}} -\Psi_2^*(-v(y)) d\nu(y) \right], \quad (11)$$

where the $c$-transform of $v$ is defined as $v^c(x) = \inf_{y \in \mathcal{Y}} (c(x, y) - v(y))$. Here, $v^c$ corresponds to the optimal $u$ given $v$.

Finally, the relationship between the marginals of the optimal UOT plan $\pi^{u,\star}$ and the original source and target distributions can be expressed using the optimal UOT potential $v^\star$ from the semi-dual problem:

**Theorem A.1** ([7, 13, 32]). *Let $v^\star$ be a solution of the dual formulation of the UOT problem between the source distribution $\mu$ and the target distribution $\nu$. Then, the marginal distributions of the optimal UOT plan $\pi^{u,\star}$ satisfy*

$$d\pi_0^{u,\star}(x) = \Psi_1^{*\prime}(-v^{\star c}(x))d\mu(x) \quad and \quad d\pi_1^{u,\star}(y) = \Psi_2^{*\prime}(-v^\star(y))d\nu(y) \quad (12)$$

## B Proofs of theorem

In this section, we provide the proof of our bias correction theorem (Theorem **??**) from the main text. Our proof builds on three key lemmas for the standard flow matching model, originally established in [22, 31], which we restate here for completeness.

**Lemma B.1** ([31], Theorem 3.1). *The marginal vector field $u_t$ generates the probability path $p_t(\boldsymbol{x}_t)$ from initial conditions $p_0(\boldsymbol{x}_0)$.*

$$p_t(\boldsymbol{x}_t) = \int p_t(\boldsymbol{x}_t \mid \boldsymbol{z})\pi(\boldsymbol{z})d\boldsymbol{z}, \quad u_t(\boldsymbol{x}_t) := \mathbb{E}_{\pi(z)} \frac{u_t(\boldsymbol{x} \mid \boldsymbol{z})p_t(\boldsymbol{x} \mid \boldsymbol{z})}{p_t(\boldsymbol{x})} = \mathbb{E}_{p_t(z | \boldsymbol{x}_t)} [u_t(\boldsymbol{x}_t \mid \boldsymbol{z})] \quad (13)$$

**Lemma B.2** ([31], Theorem 3.2). *If $p_t(\boldsymbol{x}_t) > 0$ for all $\boldsymbol{x}_t \in \mathbb{R}^d$ and $t \in [0, 1]$, then, up to a constant independent of $\theta$, $\mathcal{L}_{\mathrm{CFM}}$ (Eq. 2) and $\mathcal{L}_{\mathrm{FM}}$ (Eq.1) are equal, and hence*

$$\nabla_\theta \mathcal{L}_{\mathrm{FM}}(\theta) = \nabla_\theta \mathcal{L}_{\mathrm{CFM}}(\theta). \quad (14)$$

**Lemma B.3** ([31], Proposition 3.4). *Let the initial sample coupling be $\pi(\mathbf{z}_0, \mathbf{z}_1)$ and define the conditional vector probability path and vector field as in Eq. 3. Then, the corresponding marginal probability path $p_t(\mathbf{x}_t)$ satisfies the boundary conditions $p_0 = \pi_0 * \mathcal{N}(\mathbf{x}|0, \sigma^2)$ and $p_1 = \pi_1 * \mathcal{N}(\mathbf{x}|0, \sigma^2)$, where $*$ denotes the convolution operator. Furthermore, assuming regularity properties of $q_0, q_1$, and the optimal transport plan $\pi$, as $\sigma^2 \to 0$, the marginal path $p_t$ and field $u_t$ minimize (7), i.e., $u_t$ solves the dynamic optimal transport problem between $\pi_0$ and $\pi_1$. Specifically, $p_0 \to \pi_0$ and $p_1 \to \pi_1$ as $\sigma \to 0$.*

Here, we provide a formal statement of Theorem 3.1 and provide its proof.

**Theorem B.4** (Theorem 3.1, Formal). *Let $\pi_\tau^u$ be the optimal source-fixed UOT coupling between $\mu$ and $\nu$ with $\tau_2 = \tau > 0$ and assume that its target marginal satisfies $\nu \ll \pi_\tau^u$. Training a flow matching model with $\pi_\tau^u$ yields the biased distribution $p_1 = \pi_1^u \neq \nu$ [8]. However, applying the first-order correction (our method with $k = 1$) recovers the true target distribution $\nu$.*

$$\mathcal{L}_{\text{ours},1}(\boldsymbol{\theta}) = \mathbb{E}_{t \sim \mathcal{U}, z \sim \pi_\tau^u(z), \boldsymbol{x}_t \sim p_t(\boldsymbol{x}_t|z)} \left[ s_\tau(\boldsymbol{x}_1)^{-1} \|v_\theta(t, \boldsymbol{x}_t) - u_{t|z}(\boldsymbol{x}_t|z)\|_2^2 \right]. \tag{15}$$

*where the majority scrore $s_\tau(y)$ is defined as $s_\tau(y) := \frac{d\pi_1^{u,*}}{d\nu}(y)$. More generally, UOR-RFM with correction order $k$ generates a distribution $p_1 \propto s_\tau^{-k} \pi_{\tau,1}^u$.*

*Proof.* As an overview, the proof relies on two observations: (1) training with $\pi^u$ yields $p_1 = \pi_1^u$, i.e., the biased UOT marginal (Theorem A.1) and (2) importance reweighting with $s_\tau^{-1}$ corrects this bias, since $\nu = s_\tau^{-1} \pi_1^u$ by the Radon–Nikodym derivative. Substituting this correction into the conditional flow matching loss yields Eq. 15, and hence the generated distribution recovers $\nu$.

Formally, Lemma B.3 shows that training a flow matching model with the optimal source-fixed UOT coupling $\pi_\tau^u$, i.e.,

$$\mathcal{L}_{\text{UOT-CFM}}(\boldsymbol{\theta}) = \mathbb{E}_{t \sim \mathcal{U}, \mathbf{z} \sim \pi(\mathbf{z}), \mathbf{x}_t \sim p_t(\mathbf{x}_t|\mathbf{z})} \|v_\theta(t, \mathbf{x}_t) - u_{t|\mathbf{z}}(\mathbf{x}_t|\mathbf{z})\|_2^2. \tag{16}$$

yields a flow matching model whose boundary conditions converge to $p_0 \to \pi_{\tau,0}^u, p_1 \to \pi_{\tau,1}^u$ as $\sigma \to 0$. By Theorem A.1, we have $\pi_{\tau,0}^u = \mu$ and $\pi_{\tau,1}^u \neq \nu$. Therefore, the UOT-CFM model generates a biased distribution.

Moreover, we now show that our UOT-RFM model with the first-order correction recovers the true target distribution. From Theorem A.1, we have $\pi_\tau^u \ll \nu$, so the Radon–Nikodym derivative exists and corresponds to the majority score. By our assuption $\nu \ll \pi_\tau^u$, it follows $\nu = s_\tau^{-1} \pi_1^u$. Therefore,

$$\mathcal{L}_{\text{ours},1}(\boldsymbol{\theta}) = \mathbb{E}_{t \sim \mathcal{U}, \mathbf{z} \sim \pi_\tau^u(\mathbf{z}), \mathbf{x}_t \sim p_t(\mathbf{x}_t|\mathbf{z})} \left[ s_\tau(\mathbf{x}_1)^{-1} \|v_\theta(t, \mathbf{x}_t) - u_{t|\mathbf{z}}(\mathbf{x}_t|\mathbf{z})\|_2^2 \right]. \tag{17}$$

$$= \int_{t, \mathbf{x}_0, \mathbf{x}_1, \mathbf{x}_t} \left[ s_\tau(\mathbf{x}_1)^{-1} \|v_\theta(t, \mathbf{x}_t) - u_{t|\mathbf{z}}(\mathbf{x}_t|\mathbf{z})\|_2^2 \right] p_t(\mathbf{x}_t|\mathbf{z}) d\pi(\mathbf{x}_0, \mathbf{x}_1) dt. \tag{18}$$

$$= \mathbb{E}_{t \sim \mathcal{U}, (\mathbf{x}_0, \mathbf{x}_1) \sim s_\tau(\mathbf{x}_1)^{-1} \pi_\tau^u(\mathbf{x}_0, \mathbf{x}_1), \mathbf{x}_t \sim p_t(\mathbf{x}_t|\mathbf{z})} \left[ \|v_\theta(t, \mathbf{x}_t) - u_{t|\mathbf{z}}(\mathbf{x}_t|\mathbf{z})\|_2^2 \right]. \tag{19}$$

Note that the reweighted coupling $s_\tau(\mathbf{x}_1)^{-1} \pi_\tau^u(\mathbf{x}_0, \mathbf{x}_1)$ has the true target distribution $\nu$ as its marginal.

$$\int s_\tau(\mathbf{x}_1)^{-1} \pi_\tau^u(\mathbf{x}_0, \mathbf{x}_1) d\mathbf{x}_0 = s_\tau^{-1} \pi_1^u(\mathbf{x}_1) = \nu(\mathbf{x}_1). \tag{20}$$

Then, following a similar argument as the UOT-CFM case, our UOT-RFM model with the first-order correction recovers the true target distribution. Note that we specifically employ the sourced-fixed UOT coupling to ensure consistency with the initial conditions of the flow matching model. More generally, by a similar argument except for the normalizing constant, UOR-RFM with correction order $k$ generates a distribution $p_1 \propto s_\tau^{-k} \pi_{\tau,1}^u$. $\qquad\square$

# C  Implementation Details

This section provides the specific implementation details for our experiments on the CIFAR-10 and 2D synthetic datasets.

---

**Algorithm 1** Minibatch UOT-Reweighted Flow Matching (UOT-RFM)

---

**Input:** Empirical or samplable distributions $q_0, q_1$, bandwidth $\sigma$, batch size $b$, initial network $v_\theta$, sinkhorn target marginal weight $\tau_2$, weight power scale $k$.

$\tau_1 \leftarrow \infty$

**while** Training **do**

    Sample batches of size $b$ *i.i.d.* from the datasets: $\mathbf{x}_0 \sim q_0(\mathbf{x}_0); \quad \mathbf{x}_1 \sim q_1(\mathbf{x}_1)$

    $\pi \leftarrow \text{UOT}(\mathbf{x}_1, \mathbf{x}_0, \tau_1, \tau_2)$

    $(\mathbf{x}_0, \mathbf{x}_1) \sim \pi$

    $\mathbf{t} \sim \mathcal{U}(0, 1)$

    $\mu_t \leftarrow \mathbf{t}\mathbf{x}_1 + (1 - \mathbf{t})\mathbf{x}_0$

    $\mathbf{x} \sim \mathcal{N}(\mu_t, \sigma^2 I)$

    Calculate $\hat{s}_\tau(\mathbf{x}_1)$ from Equation (21)

    $\mathcal{L}_{Ours}(\theta) \leftarrow \hat{s}_\tau(\mathbf{x}_1)^{-k}\|v_\theta(\mathbf{t}, \mathbf{x}) - (\mathbf{x}_1 - \mathbf{x}_0)\|^2$

    $\theta \leftarrow \text{Update}(\theta, \nabla_\theta \mathcal{L}_{Ours}(\theta))$

**end while**

**return** $v_\theta$

---

**Minibatch OT Approximation** Following mini-batch OT approaches [26, 31], we approximate the UOT coupling $\pi^u$ using a mini-batch formulation similar to [10]. In practice, we adopt the POT library [11] to compute mini-batch UOT with entropic regularization [5, 12]. Specifically, for each mini-batch of training data $(\{\mathbf{x}_0^i\}_{i=1}^B, \{\mathbf{x}_1^j\}_{i=1}^B)$, the mini-batch coupling $\hat{\pi}^u$ is computed between empirical measures $\hat{\mu} = \frac{1}{B}\sum_i \delta_{\mathbf{x}_0^i}$ and $\hat{\nu} = \frac{1}{B}\sum_j \delta_{\mathbf{x}_1^j}$. Based on this, the majority score is estimated by the probability mass ratio:

$$\hat{s}_\tau(\mathbf{x}_1^j) := \frac{\hat{\pi}_1^u}{\hat{\nu}}(\mathbf{x}_1^j) = B\hat{\pi}_1^u(\mathbf{x}_1^j). \tag{21}$$

## C.1 Experiments on CIFAR-10

**Datasets** We use two datasets for our image generation experiments: the standard CIFAR-10 dataset and its long-tailed version, CIFAR-10-LT. The CIFAR-10-LT is generated to simulate class imbalance, following an exponential decay distribution. The number of samples $N_i$ for each class $i$ is determined by the formula $N_i = \lfloor N_{\max} \cdot \mathcal{I}^{\frac{i}{C-1}} \rfloor$, where $C = 10$ is the total number of classes, $N_{\max}$ is the number of samples in the largest class, and the imbalance factor $\mathcal{I}$ is set to 0.01.

**Network Architecture** We employ the U-Net architecture provided in the `torchcfm`[31], without any modifications. The architecture uses four resolution levels with two residual blocks per level in both encoder and decoder, linked by skip connections at matching scales. Each block uses $3\times3$ convolutions with Group Normalization, SiLU activations, and dropout. Down-sampling is performed by stride-2 convolutions, and up-sampling uses nearest-neighbor interpolation followed by a $3\times3$ convolution.

**Training Details** All experiments on CIFAR-10 follow the default settings of `torchcfm`. We use the `dopri5` ODE solver. For optimization, we use the Adam optimizer with a learning rate of $2 \times 10^{-4}$. The model is trained for a total of 400,000 iterations with a batch size of 128. Data preprocessing includes `transforms.RandomHorizontalFlip()` and normalization of pixel values to the range $[-1, 1]$ using `transforms.Normalize(mean=[0.5, 0.5, 0.5], std=[0.5, 0.5, 0.5])`. For stable training, we apply a warmup schedule for the first 5,000 iterations, linearly increasing the learning rate from 0 to its target value, and use gradient clipping with an L2-norm threshold of 1.0. For Unbalanced Optimal Transport (UOT), the entropy regularization parameter $\epsilon$ is set to $5 \times 10^{-2}$, and the source marginal relaxation weight $\tau_1$ is set to infinity.

**Method Details** The training process of our proposed method is as follows: (1) Sample mini-batches from each distribution. (2) Compute the coupling (transport plan) between the two mini-batches. (3) Determine the weight for each sample based on the computed transport plan. (4) Estimate the vector fields by feeding the coupled sample pairs into the U-Net and compute the weighted loss. (5) Update the network parameters via backpropagation. The specifics of each coupling method are as follows:

- **ICFM:** Uses an independent coupling, assuming the two distributions are independent.
- **OT-CFM:** Computes the transport plan $\pi$ using the `pot.emd` function and samples pairs according to the normalized probability distribution.
- **UOT-CFM:** Computes the transport plan $\pi$ using the `pot.unbalanced.sinkhorn_knopp_unbalanced` function and samples pairs based on the normalized probabilities.
- **UOT-WFM:** Also uses `pot.unbalanced.sinkhorn_knopp_unbalanced`, but samples only one target for each source sample from the normalized transport plan $\pi$.

The sample weights are calculated using the column sums of the transport plan $\pi$, which corresponds to the empirical measure of the target distribution, denoted as $\tilde{\nu}$. The weight $w(\mathbf{x}_1)$ is defined as $(1/\tilde{\nu}_{\mathbf{x}_1})^\gamma$, where $\gamma$ denotes a power factor and $\tilde{\nu}_{\mathbf{x}_1}$ denotes marginal density corresponding to a target sample $\mathbf{x}_1$. The final loss function is the weighted mean squared error (MSE) between the vector fields: $\mathbb{E}_{(\mathbf{x}_0, \mathbf{x}_1) \sim \pi} \left[ w(\mathbf{x}_1) \cdot \| v_t(\mathbf{x}_0, \mathbf{x}_1) - u_t(\mathbf{x}_0, \mathbf{x}_1) \|^2 \right]$.

**Evaluation Metrics** To assess the quality of the generated images, we use the Fréchet Inception Distance (FID), Precision, and Recall. FID scores are calculated using the `cleanfid` library. For evaluation against the standard CIFAR-10 dataset, we use the library's built-in feature statistics. For CIFAR-10-LT, the real data statistics are computed from a long-tailed dataset generated in the same manner as the training set. Precision and Recall are measured based on a widely-used implementation[1], where the real data distribution is also generated identically to the training setup.

# D   Additional Ablation Studies

Table 3: Ablation study on the correction order $k$ and the marginal matching strength $\tau$: Reported values are FID scores.

| $\tau_2 \backslash k$ | LT→LT | | | | LT→Balanced | | | |
|---|---|---|---|---|---|---|---|---|
| | 1.0 | 2.0 | 4.0 | 8.0 | 1.0 | 2.0 | 4.0 | 8.0 |
| 2.0 | 13.77 | 13.41 | 12.42 | 11.37 | 25.02 | 24.60 | 24.54 | 24.76 |
| 4.0 | 14.01 | 13.78 | 13.68 | 12.67 | 24.94 | 24.65 | 24.45 | 24.37 |
| 6.0 | 14.39 | 13.72 | 13.48 | 12.41 | 24.91 | 24.88 | 24.86 | 24.90 |

Table 3 presents an ablation study on the effects of the correction order $k$ and the marginal matching strength $\tau_2$. All models were trained on the CIFAR10-LT dataset. The "LT→LT" columns show FID scores measured against the CIFAR10-LT dataset itself, assessing fidelity to the training distribution. The "LT→Balanced" columns show FID scores using the class-balanced CIFAR10 dataset as a reference, evaluating the generation of a balanced distribution.

First, analyzing the LT→LT results, the task is to faithfully replicate the long-tailed training distribution. In this scenario, a clear trend emerges: performance consistently improves as the correction order $k$ increases. For any given value of $\tau_2$, a larger $k$ leads to a lower (better) FID score. For example, when $\tau_2 = 2.0$, the FID score monotonically decreases from 13.77 at $k = 1.0$ to a superior 11.37 at $k = 8.0$. This indicates that overcorrection ($k > 1$) is consistently beneficial, helping the model to more accurately estimate and represent the target long-tailed marginal distribution.

In contrast, the LT→Balanced setting reveals a more complex trade-off. Here, a smaller $\tau_2$ (e.g., 2.0) enables a strong corrective weight but diminishes the sampling probability of minor classes. Conversely, a larger $\tau_2$ (e.g., 6.0) improves the sampling of these classes but flattens the weights, reducing their corrective impact. This necessitates a higher correction order $k$ to induce overcorrection. For instance, with $\tau_2 = 4.0$, increasing $k$ from 1.0 to 8.0 improves the FID score from 24.94 to 24.37. However, excessive overcorrection can overshoot the balanced target, as seen for $\tau_2 = 2.0$, where the FID score worsens from 24.54 ($k = 4.0$) to 24.76 ($k = 8.0$).

---

[1] `https://github.com/blandocs/improved-precision-and-recall-metric-pytorch`

# E  Additional Qualitative Examples

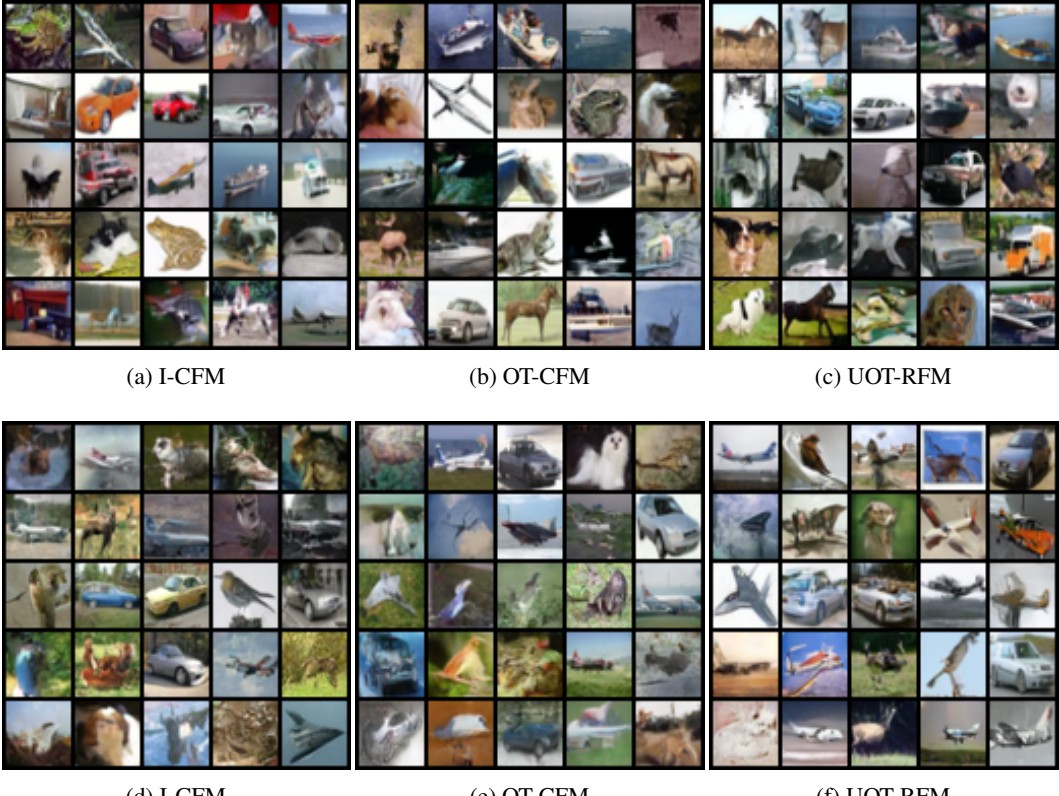

(a) I-CFM  (b) OT-CFM  (c) UOT-RFM

(d) I-CFM  (e) OT-CFM  (f) UOT-RFM

Figure 1: CIFAR image generation results: The first row shows images generated from models trained on the balanced CIFAR10 dataset. The second row shows images from models trained on the CIFAR10-LT dataset.

