# OpenReview forum: "Reweighted Flow Matching via Unbalanced Optimal Transport for Long-tailed Generation"
_NeurIPS.cc/2025/Workshop/Reliable_ML — NeurIPS 2025 - Reliable ML Workshop_

### Official Review · Reviewer_Pvka · 2025-09-18
**Decent paper, proposes a theoretically proven upsampling strategy to handle balanced data.**

**Rating:** 6
**Confidence:** 4

**Review:**

#### SUMMARY
This paper introduces UOT-Reweighted Flow Matching (UOT-RFM), a novel approach to address the significant challenge of majority bias in standard flow matching models when applied to long-tailed distributions. Standard flow matching tends to oversample majority modes and generate minority modes with low fidelity, which reduces diversity and quality for rare classes.

It derives majority score for each target data sample. This score measures how strongly each sample is emphasized by the UOT coupling. UOT-RFM corrects this bias by applying correction which is inverse weighting using the majority score.


#### STRENGTHS
* Weighting mechanism based on the majority score is entirely unsupervised, meaning it does not require class labels to identify and correct for bias
* The authors provide a bias correction theorem, formally proving that the first-order inverse weighting (k=1) with the majority score exactly recovers the true target distribution
* Proposed method is computationally efficient.

#### WEAKNESS
* Benchmarking the paper across another dataset could be useful to solidy the claim and generalizability.
* Deeper theoretical or empirical analysis for why overcorrection (k > 1) is consistently beneficial for improving fidelity (especially in LT→LT) and how to best tune it across different scenarios would significantly strengthen this aspect.
* Outlier will receive higher weight due to very low majority score and hence very high inverse weight, this could lead to undesirable effect.

#### SUGGESTIONS
* My understanding is that baching could influence the score factor for the sample, some finding around how batching could influence the score factor could help guide future research around batching strategy.

---

### Official Review · Reviewer_ewC3 · 2025-09-20

**Rating:** 6
**Confidence:** 2

**Review:**

This paper tackles the challenge of majority bias in flow matching for long-tailed generative modeling. The paper introduces UOT Reweighted Flow Matching (UOT-RFM) that can estimate an unsupervised majority score for each target data. This score is then used to reweight training samples. Experiment results demonstrate that UOT-RFM outperforms existing flow-matching baselines.

**Strengths**:
- The method comes with a theoretical bias correction result that first-order reweighting recovers the true distribution.
- Experiments show significant performance gains of the proposed method over the baselines.

**Weaknesses**:
- Only one dataset is used.
- Missing per-class or minority-focused metrics which can be helpful for assessing performance under imbalance.
- Other methods (e.g., diffusion models) can be also included as baselines.